# Effects of β-Hydroxy β-Methylbutyrate Supplementation on Working Memory and Hippocampal Long-Term Potentiation in Rodents

**DOI:** 10.3390/nu14051090

**Published:** 2022-03-05

**Authors:** Alejandro Barranco, Llenalia Garcia, Agnes Gruart, Jose Maria Delgado-Garcia, Ricardo Rueda, Maria Ramirez

**Affiliations:** 1Department of Biochemistry and Molecular Biology II, School of Pharmacy, University of Granada, 18071 Granada, Spain; abarran@ugr.es; 2SEPLIN Soluciones Estadísticas, S.L., 18004 Granada, Spain; llenaliag@seplin.es; 3Division of Neurosciences, Pablo de Olavide University, 41001 Seville, Spain; agrumas@upo.es (A.G.); jmdelgar@upo.es (J.M.D.-G.); 4Abbott Nutrition, Research and Development, 18004 Granada, Spain; ricardo.rueda@abbott.com

**Keywords:** aging, nutrition, β-hydroxy β-methylbutyrate, delayed matching-to-position, long-term potentiation, cognition, IntelliCages^®^

## Abstract

β-hydroxy β-methylbutyrate (HMB), a metabolite of the essential amino acid leucine, has been shown to preserve muscle mass and strength during aging. The signaling mechanism by which HMB elicits its favorable effects on protein metabolism in skeletal muscle is also preserved in the brain. However, there are only a few studies, all at relatively high doses, addressing the effect of HMB supplementation on cognition. This study evaluated the effects of different doses of HMB on the potentiation of hippocampal synapses following the experimental induction of long-term potentiation (LTP) in the hippocampus of behaving rats, as well as on working memory test (delayed matching-to-position, DMTP) in mice. HMB doses in rats were 225 (low), 450 (medium), and 900 (high) mg/kg body weight/day and were double in mice. Rats who received medium or high HMB doses improved LTP, suggesting that HMB administration enhances mechanisms related to neuronal plasticity. In the DMTP test, mice that received any of the tested doses of HMB performed better than the control group in the overall test with particularities depending on the dose and the task phase.

## 1. Introduction

Cognitive decline is a normal process of aging, with some cognitive abilities, such as conceptual reasoning, processing speed, and memory, being more susceptible to this gradual decline [1,2,3]. Although non-modifiable factors, like age, race and ethnicity, gender, and genetics are involved in this cognitive decline, several life-style interventions, including physical activities, cognitive training, and nutritional interventions, have been shown to have a positive impact [4,5]. In fact, several studies have shown that maintaining a healthy diet could be associated with slower cognitive decline and reduced risk of Alzheimer’s disease [6].

HMB supplementation preserves muscle mass and strength during aging by affecting the balance between protein synthesis and degradation [7,8,9,10]. The mechanism by which HMB elicits its effects on protein synthesis in skeletal muscle is related to the activation of signaling pathways involving phosphoinositide-3-kinase (PI3K), mitogen-activated protein kinases (MAPK), extracellular signal-regulated kinases (ERK1), and mTOR (mechanistic target of rapamycin) [11,12,13]. These mechanisms of action are present in the brain and are involved in brain physiology and pathology [14].

Despite the hydrophilic properties of HMB, it seems to be able to cross the blood brain barrier because it was detected in brain microdialysates after an acute oral administration in rats [15]. To add to this, a recent publication showed that HMB is transported through brain endothelial cells by a proton-coupled transport system [16]. Moreover, its efficacy has been shown in experimental models as follows: (1) it promoted the differentiation and neurite outgrowth of Neuro2a cells [17], (2) it prevented the age-related regression of dendritic trees in pyramidal neurons of rats [18], and also (3) it ameliorated age-related decline in cognitive performance in rats [19,20].

However, these studies used either in vitro doses, which have to be extrapolated to in vivo oral doses, or a relatively high nominal dose of calcium (HMB ~900 mg/kg BW of CaHMB or ~770 mg/kg BW of HMB in experimental models). The most studied doses of HMB in humans with regards to body composition and muscle strength have been 1.5 to 3 g per day in the form of calcium salt [21]. The doses in experimental animal models that showed a benefit of HMB on body composition ranged from 100 to 500 mg/kg BW of CaHMB [22]. Therefore, it is not known if these lower doses will be able to influence brain functionality.

The aims of this study were: (1) to reproduce the effect of HMB on brain functionality using an automated system designed for behavioral studies in rodents (IntelliCage^®^, NewBehavior AG, Zurich, Switzerland) [23] using a behavioral task to assess the spatial working memory in animals (DMTP task) [24] (2) to test if HMB was able to affect LTP, an electrophysiological mechanism underlying synaptic plasticity and memory storage [25], and (3) to study the effective dose of HMB on brain function in adult and middle-aged rodents.

The doses were selected based on the literature and body weight and body surface of animals, and its relationship with human equivalent dose (HED) [26]. Because the conversion factor for mice was twice that of rats (12.3 vs. 6.2), the doses used in both experiments were similar when calculated as HED, and were approximately between 2 and 9 g of HMB per day for an average 60 kg body weight in humans.

## 2. Materials and Methods

### 2.1. Animals

Male C57/BL6 mice (*n* = 48; 48 week-old) and male Long-Evans rats (*n* = 60; 32 week-old) were purchased from Janvier (Janvier, Saint-Berthevin Cedex, France) and kept under controlled environmental conditions of temperature (22 °C ± 2), humidity (55% ± 10), and lighting (12 h light/dark cycles).

The experiments were conducted in mice and rats for DMTP and LTP, respectively, because the equipment and experimental techniques were adapted to these species: Intel-liCages are optimized for mice and the implantation of electrodes for LTP is easier in rats than in mice due to the size of the skull.

Animals were fed a standard diet ad libitum. Animal experimental protocols were approved by the Ethics Committee of the Estación Experimental del Zaidín-CSIC (Granada, Spain. Approval number: CBA EEZ-2011/20), and the experiment was performed in accordance with the Spanish and European regulations for the care and use of experimental animals for research.

### 2.2. Experimental Designs

#### 2.2.1. Long-Term Potentiation Study in Rats

The rats were housed individually in standard cages with free access to food and water until they were 52 weeks old. Animals were divided in four study groups; each group received a specific HMB dose (D): D1 = 225, D2 = 450 and D3 = 900 mg/kg·BW/day (*n* = 15/group). Calcium HMB was dissolved in gelatin to facilitate the intake. The gelatins were offered twice a day for 2 months. Rats in the control group (C) were given equivalent doses of Ca as Ca-Lactate. The LTP measurements were done when the rats were 62–68 weeks old (15.5–17 months old). The administration of CaHMB continued during the procedure; in total the rats received the supplement for 4 months.

Animals were anesthetized with 4% chloral hydrate at a dose of 1 mL/100 g. Once anesthetized, animals were implanted with stimulating and recording electrodes in the hippocampus. Stereotaxic coordinates [27] were followed to implant animals with stimulating electrodes aimed at the Schaffer collateral-commissural pathway of the dorsal hippocampus (3.5 mm lateral, 3.2 mm posterior, and 3.0 below to Bregma). These electrodes were made of 50 µm, Teflon-coated tungsten wires (Advent Research Materials Ltd., Oakfield Industrial State Eynsham, Oxford, England). In addition, animals were implanted with recording electrodes aimed at the ipsilateral stratum radiatum underneath the CA1 area (2.5 mm lateral, 3.6 mm posterior, and 2.3 below to Bregma). Recording electrodes were also made of 50 µm Teflon-coated tungsten wires (Advent Research Materials Ltd., Oakfield Industrial State Eynsham, England). Electrodes were surgically implanted in the CA1 area using as a guide the field potential depth profile evoked by paired (20–50 ms of interval) pulses presented to the ipsilateral Schaffer collateral pathway. Recording electrodes were fixed at the site where a reliable monosynaptic field excitatory post-synaptic potential (fEPSP) was recorded. A 0.1 mm bare silver wire was affixed to the skull as a ground. All wires were connected to two separate sockets (RS-Amidata, Madrid, Spain). The ground wire was also connected to the recording system with a single wire. Sockets were fixed to the skull with the help of two small screws and dental cement [28,29,30].

Recordings were carried out using Grass P511 differential amplifiers within a bandwidth of 0.1 Hz–10 kHz (Grass-Telefactor, West Warwick, RI, USA). Synaptic field potentials in the CA1 area were evoked by single 100 µs, square, biphasic (negative-positive) pulses applied to ipsilateral Schaffer collaterals. Stimulus intensities ranged from 50 to 350 µA. For each animal, the stimulus intensity was set well below the threshold for evoking a population spike, usually 30–40% of the intensity necessary for evoking a maximum fEPSP. An additional criterion for selecting stimulus intensity was that a second stimulus, presenting 20–50 ms after a conditioning pulse, evoked a larger (>20%) synaptic field potential [28,29].

For evoking LTP in alert behaving rats, we used the following HFS protocol: each animal was presented with five 200 Hz, 100 ms trains of pulses at a rate of 1/s. These trains were presented six times in total, at intervals of 1 min. The 100 µs, square, biphasic pulses used to evoke LTP were applied at the same intensity used for evoking baseline records [29,31]. Before the HFS protocol, baseline records were collected for 15 min with the paired stimuli presented every 20 s. After the HFS protocol, fEPSPs were recorded again for 60 min. Additional recordings were carried out for 30 min during the 3 following days.

For LTP, collected data were represented in Excel sheets for further analysis. Only those rats that completed the electrophysiological study and electrodes located in the selected stimulating (CA3 area) and recording (CA1 area) were considered for the quanti-tative analysis of fEPSP evoked during the LTP study. Unless otherwise indicated, data are presented as the mean value collected from each experimental group followed by the S.E.M. Graphic displays were constructed with the help of the SPSS package (SPSS Inc, Chicago, IL, USA). Statistical differences between experimental groups and their corresponding control were determined with the help of the same statistical package. Data were analyzed using a Two-Way Repeated Measures ANOVA test, with time as repeated measure, followed by All Pairwise Multiple Comparison Procedures (Fisher LSD Method). Statistical procedures and other analytical details have been described elsewhere [28,32].

#### 2.2.2. Delayed Matching-to-Position Task in Mice

Mice were maintained at the animal house for four weeks and then, divided into four groups (*n* = 12 per group) matched by body weight. They were adapted to the IntelliCages© system and trained to drink water twice daily for two weeks. Calcium HMB was dissolved in water and administered in the drinking bottles. Three doses of HMB were used: Dose 1 = 450, Dose 2 = 900, and Dose 3 = 1800 mg/kg·BW/ day. Mice in the control group were given water with equivalent doses of calcium as Ca-Lactate. This procedure allowed us to provide HMB twice a day and to control the dose. After one month, spatial working memory was evaluated. The procedure lasted for 6 weeks and during the cognitive evaluation mice also received the supplementation; in total, they received the supplement for 10 weeks. The mice were 59–65 weeks of age (15–16 months) at the time of evaluation.

The IntelliCage© (NewBehavior AG, Zurich, Switzerland) is a computer-based, fully-automated testing apparatus used to analyze the spontaneous and learning behavior of rodents. This system consists of a cage that presents four operant conditioning corners, which can locate one mouse at a time. Each corner is equipped with two motorized doors, which block or allow access to water bottles placed on both sides of the corner [33]. When a mouse tries to access through whichever of the two doors (nose poke action), the interruption of a light-beam sensor at either door triggers one or the other doors to open and allow access to a water bottle. Three mice were allocated per corner to reduce competence for water intake. This way the variability of water intake and HMB dose was minimized. Radio frequency identification (RFID) transponders are implanted under the mouse skin, allowing for individual recognition; thus, mouse entries into the corners are detected through RFID antennas located there [34].

A RFID-transponder was injected subcutaneously in the interscapular area of each mouse. After transponder implantation, the mice were placed into the IntelliCages© at the beginning of the dark phase and maintained there during two weeks for habituation. The habituation process consisted of different stages: one day of free exploration with all doors opened; five days in which doors were opened only upon a visit to the corner; in the following two days, doors opened after a nose poke in the right place; in the last four days, one nose poke opened any door only during two drinking sessions of 90 min per night, while the doors remained closed during the rest of the night and day (adaptation to IntelliCage is showed in Figure 1).

Mice were evaluated with the DMTP test for assessing spatial working memory. The protocol consisted of two phases. First, pre-training: mice acquired the matching rule. Mice were trained to NP (nose-poke) either the left or the right door on successive trials. Each NP was rewarded with access to water bottle for 4 s.

The illumination of the single door with the first NP (matching door) at the beginning of a trial constituted the sample phase; the illumination of both doors occurred 3 s after lighting the first door and constituted the choice phase. The choice was presented immediately after the NP without retention delay (0 s). Mice were rewarded for choosing the first door that was nose-poked during the sample phase. Any incorrect (error) response resulted in the end of the test, in which the lights were extinguished (Figure 2a).

The second phase: DMTP training was identical to the choice phase except that a selected retention delay interval was interposed between the sample and choice phases of the task. The first NP at the matching door after the delay interval results in the opening of the door. The delay time was chosen to be 0, 1, 2, 4, and 8 s (Figure 2b). A subject received up to 50 trials with reward over the course of each session. Typically, there is a learning-curve over time for each experimental stage of DMTP starting with low success-proportions, increased performance over time, and stabilization phase or plateau. The plateau indicates that learning at this level is finished. The pass to the next level was decided by fitting linear mixed models to the percentages of correct results every day and the last five days before. If the slope of the line for the four groups was not significantly different from 0 at *p* < 0.05, then the stability of the learning curve within the delay was reached and the delay was changed to the next step.

To address the question about differences in the DMTP learning process among groups, a generalized additive mixed model (GAMM) with binomial distribution to the (raw observations) success-data of the mice using the logit link function was applied. Note that mixed models allow incorporating repeated measures from individual mice by accounting for the within-subject correlation. GAMMs allow fitting a smooth curve over time by group of treatment to compare the progression of the learning process. GAMMs are applied to data from each delay over days. The model accounts for all data from all days of that delay with two main effects group and the smooth curve of response over days by groups. For each delay, a final group comparison based on the last day of each experimental delay period was performed using GLMM (Generalized Linear Mixed Model) for binomial data with the logit link function, as we assume here that learning has reached the final stage. Group differences were tested by fitting the GLMMs for each delay. Furthermore, to test for differences between delays depending on the group, an accomplished complete model taking observed behavioral data for all mice each time on the final day of delay (T = 0, 1, 2, 4, and 8 s) was fitted with delay, group, and interaction effect.

After the last training period (8 s), a retraining with delay T = 1 s was performed to compare training and retraining success proportions as well as to compare the different groups at the retraining stage. A GAMM with logit-link function was employed. The main effects of group and stage (retraining and training), as well as the interaction, were considered. All tests were performed at a 5% significance level. The R-software was used to perform the statistical analysis.

## 3. Results

### 3.1. HMB Effects on Hippocampal LTP Evoked in Alert Behaving Rats

The results collected from the LTP study are illustrated in Figure 3. The four groups presented a significant LTP during the first recording session after the HFS protocol in comparison with baseline values (Day 1). Both medium (D2 = 450 mg/kg. BW) and high dose groups (D3 = 900 mg/kg. BW) presented a larger and longer lasting LTP than the other two groups (C = Control, and low dose (D1 = 225 mg/kg.bw). Although the D2 group reached larger LTP values than the D3 group, mainly during the first recording session after the HFS protocol, no significant differences could be observed between them.

### 3.2. HMB Effects on Delayed Matching to Position Task in Mice

#### 3.2.1. Acquisition of the Delayed Spatial Matching-to-Position Task

Figure 4 shows the adjusted curve of probabilities of correct results by treatment group (left graph), for delay T = 0. There were no differences between groups on the day when the stability of learning curve was reached. However, Dose 1 and 3 groups learnt faster than the other two groups during the exponential learning phase (on days 2 to 6, D1 and D3 > C, significant differences are shown in Figure 4, right panel). Dose 2 performed similar to controls during the first four days but surpassed control group after day 5, reaching a closer performance to the other doses.

#### 3.2.2. Intervals Test of the Delayed Spatial Matching-to-Position Task

Figure 5 shows the observed percentage of success for the different groups and delays. The vertical dash lines represent the change of delay (delay intervals of 1, 2, 4, or 8 s). It was observed that the percentage of correct visits in the stability phase of learning decreased with delays in all groups due to the increased difficulty of the task.

Dose 2 appeared to be superior compared with the other three groups, except for the last 2 days of delay T = 8. The control group had the lowest percentage of correct visits.

To confirm this difference, a GAMM was fitted to the data. Figure 6 shows the overall fit and the statistic test for comparing the adjusted average probability of success over days at 5% significant level. The mice that received any dose of HMB performed better than the control group (*p* < 0.05). Moreover, the animals that received Dose 2 significantly improved their success rate compared with the others HMB groups.

#### 3.2.3. Retraining Effects in the Delayed Spatial Matching-to-Position Task

After completing delay T = 8, a retraining phase with a delay T = 1 was carried out to analyze if retraining has some effect on learning. A GAMM was fitted to all days of the train and retrain process adjusted by group, stage (train/retrain), and interaction between them, as well as a smooth curve over days by group/stage. The comparison based on estimated success-probabilities with corresponding 95%-confidence intervals is shown in Figure 7 organized by groups (a) and by training-stage (b). As it can be seen in the left panel, a significant improvement of retraining over training was achieved by all groups. During the first training, only the Dose 2 group was significantly different from the control, while during retraining all the treatment groups had significantly higher probabilities of success than the control group.

## 4. Discussion

In the current study, the effect of HMB supplementation at different doses on cognitive skills and electrophysiology measurements in middle-aging rodents was evaluated. We assessed cognitive functions by an adaptation of the classic DMTP in mice to the Intellicages© system. The cognitive assessments were complemented by electrophysiological measurements during the experimental induction of LTP at the hippocampal CA3-CA1 synapse in rats. The results presented in this study corroborate that oral supplementation with HMB improves cognitive functions and provide insights about the effective dose of HMB on brain functionality.

The present work uses a totally automated system (IntelliCage©, NewBehavior AG, Zurich, Switzerland) for the behavioral study. The IntelliCage^®^ is designed for behavioral studies that allows the assessment of animals in group, but with an individual profiling [23]. Most of the home-cage systems currently available for behavioral monitoring require to house mice in isolation [35]. However, it is showed that social isolation induced depression-like behavior in mice [36]. In addition, the handling and/or placement of mice in novel arenas and mazes for testing causes stress and changes in behavior. Therefore, the monitoring of mice in a home-cage environment and the absence of human interference during behavioral assessment could allow a greater degree of standardization of cognitive tests in mice between laboratories [37]. DMTP is one of the most frequently employed behavioral tasks for assessing spatial working memory in animals [38]. As far as we know, this is the first time that the DMTP test in mice has been adapted to the IntelliCage^®^ system, and the protocol described here could be used in future research using this system.

Animal cognitive assessments were complemented by electrophysiological data based on LTP measurements carried out in alert behaving rats. LTP is an experimentally evoked process, whereby synaptic strength is rapidly increased, and the modification of synaptic strength produced by LTP is widely thought to underlie memory storage [25].

Larger and longer lasting LTP were observed in rats supplemented with the medium and high dose of HMB than in rats of the control group. However, the performance of animals supplemented with the low dose was not different from control group. These data suggest that a certain HMB concentration threshold should most likely be exceeded to observe changes of LTP.

In the DMTP task, all the doses of HMB were somehow effective depending on the task. In the acquisition phase (no delay, T = 0), mice learnt to associate a NP with water reward. The task was quite difficult for the animals because mice needed 8 days to reach the maximum learning level. In fact, during the first days the percentage of correct results was very low for all mice, and it only started to improve after four or five days. This result contrasts with other studies, where the mice learned within the first two days of training [24,39]. This different performance could be due to the differences between Intellicage© design and the Skinner boxes usually used for the classical version of the test [40].

Mice supplemented with a low and medium dose of HMB learned the task better than the control the first day. This result suggests that HMB was able to give a cognitive advantage when the mice faced a complicated task for the first time. A very interesting effect on learning speed was also detected. Mice supplemented with high and low doses of HMB were the fastest learners. These results suggest that the cognitive effect of HMB may follow a U-shaped dose-response curve for some cognitive abilities. However, when the task was analyzed as a whole (Figure 6), all the doses of HMB improved the probability of success in comparison to controls, indicating that HMB was effective at all the doses given.

During aging, some cognitive tasks that require speed of processing, executive function and working memory could be altered [1,2]. Retraining and practice may help the adjustment and improve performance [41]. Moreover, some cognitive interventions for dementia [42] or early rehabilitation [43] included cognitive retraining to enhance cognitive performance. Cognitive retraining in research animals has not been widely studied. In our opinion, it is a very interesting phenomenon to explore for nutritional interventions, given the potential in the care of cognitive health. The present study showed that a previous cognitive training in DMTP task facilitated the retraining performance. All groups, including the control, performed better after a retraining process. However, unlike the training stage, where animals supplemented with medium dose of HMB learned significantly better than the rest of the groups, in the case of retraining all the supplemented groups performed better than the control animals. Moreover, supplemented animals with the highest dose of HMB learned better than the other groups (low and medium). This result suggests that HMB could also be involved in the long-term memory process. Moreover, this process seems to be more optimal at a high HMB dose or could need a longer supplementation period.

Our results contrasts with a previous study by Munroe et al. [44] but agree with those of Kougias, et al. [19] and Hankosky et al. [20]. Munroe et al. did not find differences in cognitive performance between mice supplemented with HMB at 450 mg/kg BW and controls. This discrepancy could be due to a longer time of supplementation in our study (10 vs. 5.5 weeks) or to the type of learning analyzed. They analyzed fear-conditioned learning (passive avoidance) and recognition-based learning (novel object recognition). We analyzed spatial working memory in DMTP task. It was reported that DMTP test is a sensitive assay of hippocampal function [45]. In fact, most of the studies in experimental animals showed that damage to the hippocampus results in delay-dependent impairments on this task, indicating an impairment in working memory [46]. Our results on hippocampal LTP in rats supported that the positive cognitive effects of HMB could be mediated by an increase of synaptic strengths, or complexity to the synapse diversity, evidenced by an increased LTP [47]. With regards to Kougias, et al. [19] and Hankosky, et al. [20], we showed that HMB affected cognitive performance at lower doses than the one used in these two studies, and within the range of the studies showing efficacy on body composition and muscle performance.

There is only one study in humans that has evaluated a supplement containing HMB on cognitive performance with or without exercise in active-duty air force men. The supplement was designed to support both muscle and cognitive performance and contained not only HMB (3 g/day CaHMB) but also other nutrients able to affect cognitive performance such as B vitamins, lutein, and DHA. The nutritional intervention improved working memory, fluid intelligence reaction time, and processing efficiency [48]. Although it was a multi-nutrient intervention, taking into consideration our results, HMB may be well considered a contributor to these positive cognitive results, beyond its efficacy on muscle health.

## 5. Conclusions

Overall, our findings establish the beneficial effects of oral HMB supplementation on cognitive function and electrophysiological measurements in adult and middle aged rodents. The results from this study suggest that some of the effects may be dose-dependent but overall, all the doses were efficacious in one task or more. Nevertheless, this study showed that HMB at the dose typically used for body composition and muscle performance improved brain functionality, namely working memory, which is an important function to be preserved during aging. However, more studies in humans are needed to translate these results into the clinical setting. The inclusion of cognitive evaluation in future clinical studies using HMB supplementation is encouraged.

## Figures and Tables

**Figure 1 nutrients-14-01090-f001:**
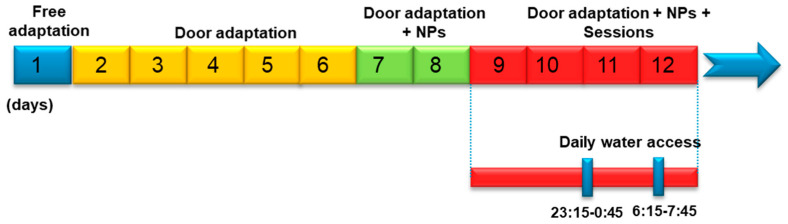
Mice adaptation to IntelliCages Timeline: Free exploration: all doors of the IntelliCage were opened; Door adaptation: doors were opened after a corner visit; Door adaptation + NP: the respective doors were opened after a NP response during a corner visit; Door adaptation + NP + Sessions: one NP opened the respective door, once during a visit and during two drinking sessions per night when water was available. The doors remained closed during the rest of the time.

**Figure 2 nutrients-14-01090-f002:**
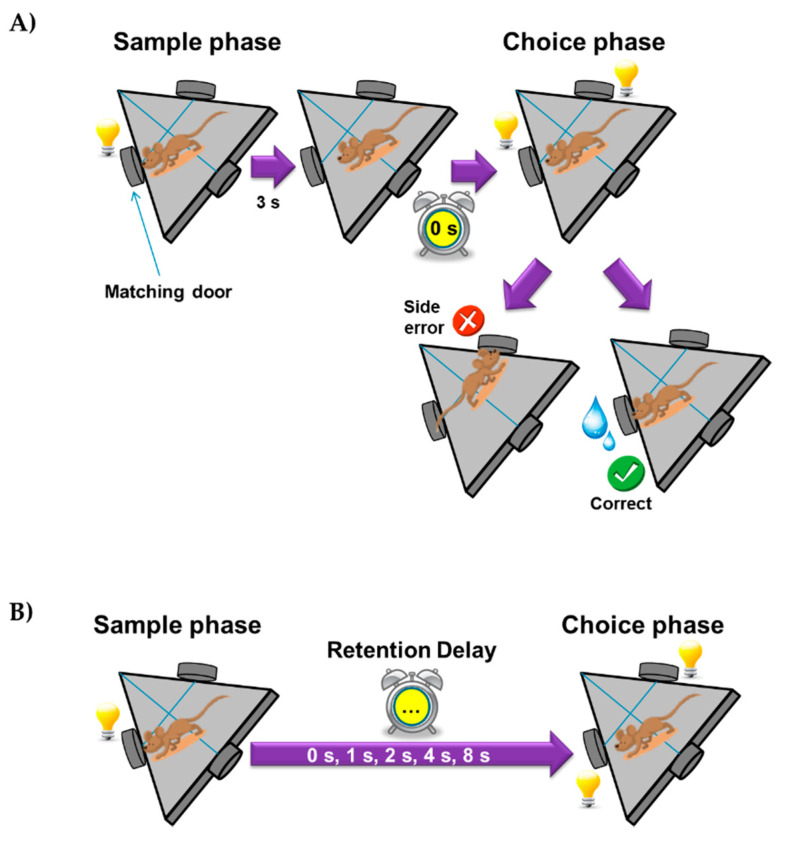
DMTP procedure: (**A**) pre-training, (**B**) training.

**Figure 3 nutrients-14-01090-f003:**
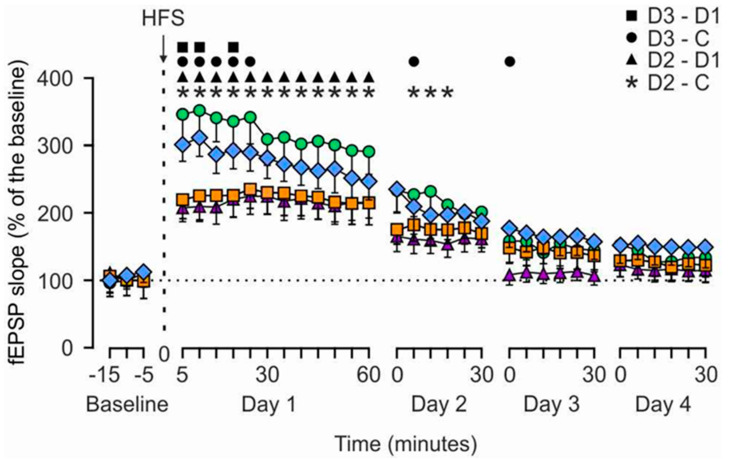
Results of LTP evoked in the four (Control, D1, D2, D3) rat groups. Animals were stimulated with the high-frequency stimulation (HFS) protocol described in the Methods section. Recordings were carried out for 15 min before to 60 min after the HFS protocol (Day 1). Additional recordings were carried out for 30 min during 3 additional days (Days 2–4). Data were collected from *n* = 16 electrodes/group implanted in *n* ≥ 5 animals/group. The statistical analysis (Two-way Repeated Measures ANOVA, F(96, 1920) = 2.818; *p* < 0.001) indicated that the four groups reached significant LTP with respect to baseline values for days 1 to 3 (D2 and D3; *p* ≤ 0.05) and days 1 and 2 (Control and D1 groups; *p* ≤ 0.05). In addition, the All Pairwise Multiple Comparison Procedure (Fisher LSD Method) confirmed that the D2 (green line) group presented significantly larger LTP values than the Control (violet line; *, *p* ≤ 0.05) and the D1 (orange line; ▲, *p* ≤ 0.05) groups on days 1 and 2. In addition, the D3 group (blue line) also presented significantly larger LTP values than the Control (●, *p* ≤ 0.05) and the D1 (■, *p* ≤ 0.05) groups at the indicated times on days 1 to 3. No significant differences were observed between C and D1 groups (*p* ≥ 0.671) or between the D2 and the D3 groups (*p* ≥ 0.198). C = Control, D1 = 225 mg/kg. BW, D2 = 450 mg/kg. BW and D3 = 900 mg/kg. BW HMB.

**Figure 4 nutrients-14-01090-f004:**
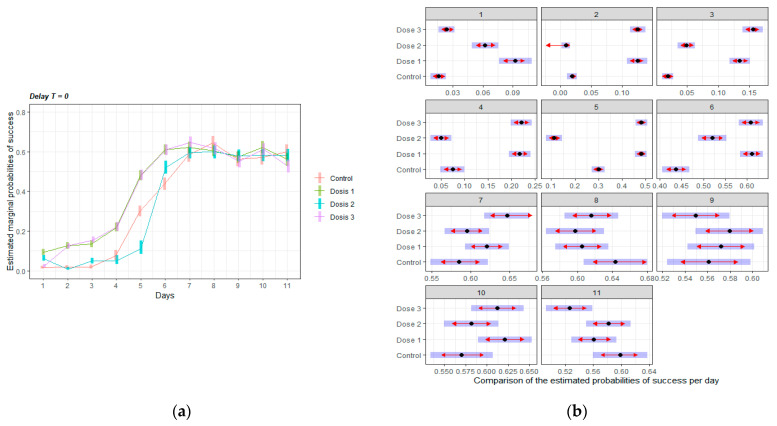
Learning curves of mice in the DMTP task without delay: (**a**) Adjusted curve of probabilities of correct results by treatment group, for delay T = 0. (**b**) Marginal estimate from the model for each day and treatment, the blue shadows represent the 95% confidence interval, and the red arrows represent testing differences between groups. If the red arrows from two different groups overlap, no significant differences are found at 5% significant level. If the red arrows from two different groups are not overlapping, then a significant difference is found at the 5% significant level. Dose 1 = 450, Dose 2 = 900 and Dose 3 = 1800 mg/kg·BW HMB.

**Figure 5 nutrients-14-01090-f005:**
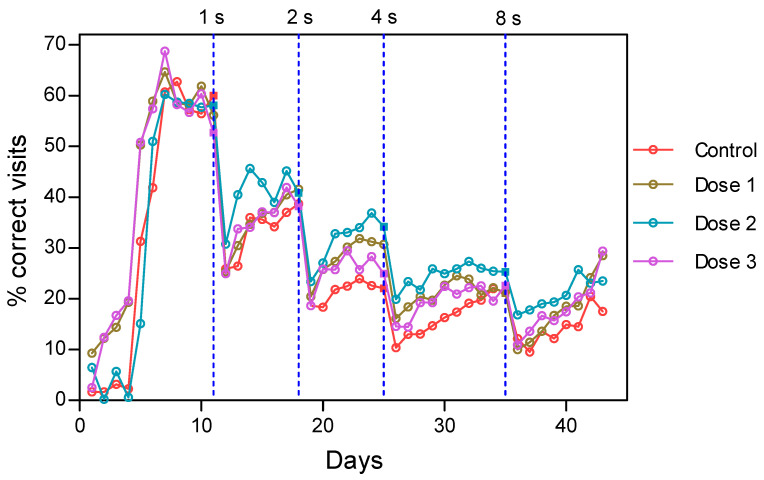
Percentage of success of the DMTP task for the different groups and delays. The vertical dash lines represent the change of delay. Dose 1 = 450, dose 2 = 900, and dose 3 = 1800 mg/kg·BW HMB.

**Figure 6 nutrients-14-01090-f006:**
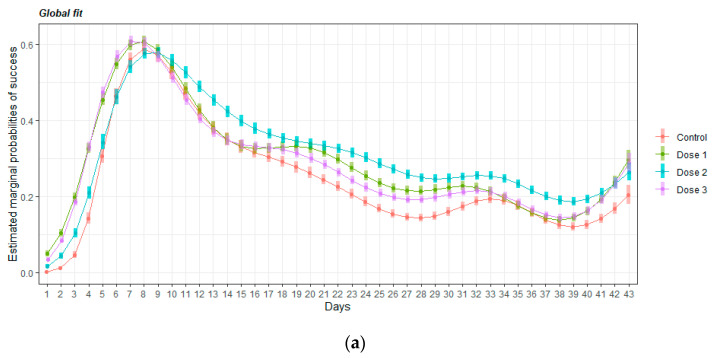
Probability of success calculated on the DMTP task for the different groups and delays. (**a**) Overall fit for the adjusted average probability of success over days; (**b**) marginal estimate from the model for each day and treatment, the blue shadows represent the 95% confidence interval, and the red arrows represent testing differences between groups. If the red arrows from two different groups overlap, no significant differences are found at 5% significant level. If the red arrows from two different groups are not overlapping, then a significant difference is found at the 5% significant level. Dose 1 = 450, dose 2 = 900, and dose 3 = 1800 mg/kg·BW HMB.

**Figure 7 nutrients-14-01090-f007:**
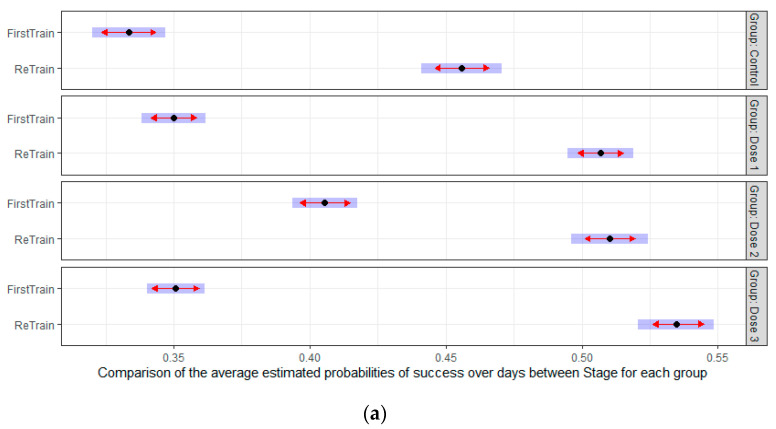
Marginal estimate from the model for the first training and retraining organized by groups (**a**) as well as by training-stage (**b**). The blue shadows represent the 95% confidence interval and the red arrow represent the testing differences between groups. If the red arrows from two different groups overlap, no significant differences are found at 5% significant level. If the red arrows from two different groups are not overlapping, then a significant difference is found at the 5% significant level. Dose 1 = 450, dose 2 = 900, and dose 3 = 1800 mg/kg·BW HMB.

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
