# Peer review of "Effects of β-Hydroxy β-Methylbutyrate Supplementation on Working Memory and Hippocampal Long-Term Potentiation in Rodents"

_nutrients, 2022, doi:10.3390/nu14051090_

Round 1
Reviewer 1 Report
General Comments
This manuscript investigates the effects of HMB at relatively low, medium, and high experimental doses on the potentiation of hippocampal synapses following experimental LTP induction in rats as well as on the working memory of mice. Although this study is unique and adds to the existing literature of HMB’s nootropic effects with some distinguishable doses, there are instances of inflated statements, deficiencies in the rationale for specific details of the study design, and minor reporting discrepancies/inconsistencies.
Additionally, extensive editing of the English language and stylistic writing would substantially enhance this paper. Specific comments and recommendations are outlined below to provide guidance in making this suitable for publication in Nutrients. For the most part, although these comments are presented below based on "major and minor comments, all comments herein are relatively minor and should not be too difficult to address.
Major Comments
• Throughout the manuscript, there are overstated claims that are beyond the scope of this paper with respect to relative dose that need to be addresse.
o L. 19: change “extremely” to “relatively”
o L. 19-20: remove “3-fold beyond acceptable supplementation dose”
- Based on what? Context is important. Unless you’re providing a PBPK model with clear indication of how this translates to humans, this is unsubstantiated, as there is no “acceptable” supplementation dose
- In fact, HMB appears to be practically non-toxic in experimental animals tested
according to OECD guidelines (see Acute dose toxicity evaluation of the food supplement calcium 3-hydroxy-3-methylbutyrate (HMB) in female Sprague Dawley rats - PubMed (nih.gov))
o L. 27-29: this statement is beyond the scope of this paper and certainly does not belong in the abstract. Perhaps if there was more context, this statement may be placed in the conclusions, with some caveats.
- Although lines 335-340 provide some justification for the doses used in this study, the authors should give further consideration to dose translation if making this
statement, as this type of allometric scaling is useful in many ways, but is not always accurate and is inferior to chemical-specific information fed into a PBPK model (seeTranslating dosages from animal models to human clinical trials--revisiting body
surface area scaling - PubMed (nih.gov))
o L. 59: add “relatively” and “nominal” and normalize to HMB daily dose, as it appears that 450 mg Ca-HMB/kg was the target dose for the two doses administered daily, with only one dose on Sunday; therefore, 450 mg Ca-HMB × 13 doses/7 days × ~85.4% (HMB:Ca-HMB)
- i.e., “a relatively high nominal dose of approximately 700 mg HMB/kg/day in the
form of a calcium salt”
o L. 58-64: This is a great paragraph that adds context; however, please normalize all doses with respect to HMB-specific doses, whether it be from free acid form of HMB or calcium HMB
• Please provide rationale for the use of male animals only
• Please provide rationale for the use of a 32-week-old rat, as a middle-aged rat
• Please provide rationale for using mice AND rats, as opposed to just one of the species
o This is only briefly touched upon in lines 332-334
• There is a lot of background information in the discussion that should appear either in the intro or the methods (e.g., l. 332-340
• Given that 12 mice were housed together, how were the doses appropriately separated for each individual mouse? If they were not, perhaps commenting on the potentially low variability of mouse size/cage and water intake/mouse?
• L. 86 (15 rats/group) vs l. 233 (n ≥ 5 rats/group): what is the reasoning for this difference?
• L. 253 seems to be missing the beginning part of a sentence and “(b)” after “Table 0”
• Figure 5: I like how the authors included the vertical dash lines to represent the delay change, but if the authors were to perhaps use a solid square or some other symbol that is easily distinguishable from the open circle for the data point representing when the stability phased was reached for each group, this would greatly improve this figure.
Minor Comments
• Extensive editing of the stylistic writing
o Choppy abstract – Here are some suggestions:
- Combine sentences in l. 15-16 of the abstract: β-hydroxy β-methylbutyrate (HMB), a metabolite of the essential amino acid leucine, has been shown to preserve muscle
mass and strength during aging - L. 16-20: The signaling mechanism by which HMB elicits its favorable effects on skeletal muscle is also preserved in the brain. However, there are only a few studies, all at
relatively high doses, addressing the effect of HMB supplementation on cognition.
o First paragraph (l. 34-42) of introduction is disjointed:
- What is the antecedent to “these possible risk factors?” (l. 42)
- Perhaps revising to the following (n.b., citations need to be transferred where
necessary):
• Cognitive decline is a normal process of aging, with some cognitive abilities, such as conceptual reasoning, processing speed and memory, being more susceptible
to this gradual decline. Although non-modifiable factors, like age, race and ethnicity, gender, and genetics, are involved in this cognitive decline, several
lifestyle interventions, including physical activities, cognitive training, and nutritional intervention, have been shown to have a positive impact.
• Extensive editing of the English language (n.b., these are just examples – please thoroughly review entire manuscript)
o Abstract (l. 17): elicits
o Intro (l. 45): elicits
o Intro (l. 49): in brain physiology
o Intro (l. 61): have instead of has
o Materials and Methods (l. 73): “male” does not need to be capitalized
o Based on the definition of the stability phase (l. 262-263), the proceeding sentence (l. 262-265) should say “was reached on days 6 and 7"
• Minor reporting deficiencies/inconsistencies:
o Intro (l. 55-56): it prevented the age-related regression of dendritic trees in pyramidalneurons of rats
o Long-Evans is hyphenated (l. 73) vs not hyphenated (l. 84)
o L. 85, 135-136: define “D” as dose (e.g., D1, D2, D3)
o Figure 3: HFS should be aligned with the dotted vertical line at 0 – perhaps add a downward arrow below HFS that points to that
o L. 86 (15 rats/group) vs l. 233 (n ≥ 5 rats/group): what is the reasoning for this difference?
o L. 218-219: “Only D2 and D3 groups still presented a significant LTP 24 h later (Day 2)”
- Where are the statistics to support this?
- I don’t see any statistics regarding day 2 in comparison to baseline nor do I see any symbols to designate that comparison in Figure 3
o L. 205 (T=0) vs l. 253 (Table 0)
- Define T... I’m presuming this should say time, not table
o L. 266: delay T=4 ... and delay T=8
o L. 268: T1 to T8 vs T=1 to T=8
o Section 3.1 (l. 215 – 227) uses C, D1, D2, & D3, while the proceeding results subsections all spell out these groups (e.g., Dose 1, Dose 2, etc.)
• L. 140: Please add the days of cognitive evaluation (total days of supplementation), as this is not abundantly clear
• Please provide rat housing conditions
• L. 395-402: provide dose of HMB in this human study – I believe it’s 3 g/day, but if this can be matched with average subject body weight to provide a dose in mg/kg/day, that’d be ideal
• Results & interpretations:
o Confirmation question re lines 218-219: So, there is no statistically significant difference between fEPSP slope on day 2 vs baseline for D1? The only reason I ask is because visually in Figure 3, I’m just surprised given the relative small SEMs and the almost 2-fold increase on Day 2 relative to baseline.
o Too much weight was given on nuanced findings that were not statistically significant (or that meaningful)
- E.g., line 269: “except for the last 2 days of delay T8” – consider removing and replacing the beginning of the sentence with “... Dose 2 generally appeared to
be superior” - E.g., lines 249-251: why is this minor, potentially spurious finding mentioned when the statistically significant differences found on Day 1 (D1 & D2 > D3 & C), Day 5 (C > D2), and Day 11 (C > D3) are not mentioned?
- E.g., line 277 “The mice that received any dose of HMB performed better than the control group” is understated and needs to be mentioned FIRST before
discussing the nuanced findings that have lesser significance/potential clinical relevance
• General comment: I find it a little odd that statistics are tucked in figure captions, as opposed to the main text.
o For example, l. 277-278: “The mice that received any dose of HMB performed better than the control group.” It is unclear that this is a statistically significant finding by itself until one looks at figure 6 caption (or figure 6 b). Simply adding “(p ≤ 0.05)” at the end of that particular sentence, and similar sentences throughout the results sections, would make it clear whether the authors were making a general observation or stating a statistically significant result.
Author Response
A detailed replay to reviewer 1 is in the attached document. The responses are in red.

Reviewer 2 Report
The article by is very interesting, as it brings a new aspect of HMB properties. Usually HMB is consumed to enhance muscle anabolism and not brain performances. 2 species are studied using 2 ways to analyze the effects, and it would be interesting to perform the electrophysiology on the mice brains as well.
The methods describe two different ways to supply the product: gel for rats v liquid for mice. Do the authors have any idea of the amount actually consumed? Is it comparable? Also, based on the doses provided, an evaluation of 2.2/4g per day for humans is quite high, re-evaluating with the actual amount ingested could reduce this figure.
The fact that mice were not in isolation is very original and interesting; just for curiosity, it would be interesting to compare with isolated animals.
Lines 182 to 205 are difficult to follow
What is the rationale for the doses? Why are the doses so high for the mice?
Author Response
Comments and Suggestions for Authors
We thank the reviewer for his/her comments. Some of them we also highlighted by reviewer 1. Here are our responses and comments.
1. The article by is very interesting, as it brings a new aspect of HMB properties. Usually HMB is consumed to enhance muscle anabolism and not brain performances. 2 species are studied using 2 ways to analyze the effects, and it would be interesting to perform the electrophysiology on the mice brains as well.
We agree with reviewer, it would have been interesting to test the two techniques in the two species or to use only one of them. However, there were some limitations. LTP was done in vivo and required the implantation of electrodes in the hippocampus. The animals performed the task in free living conditions with no restrain. The intervention was easier in rats than in mice, as well as the location of the electrodes in a precise area of the brain.
2. The methods describe two different ways to supply the product: gel for rats v liquid for mice. Do the authors have any idea of the amount actually consumed? Is it comparable? Also, based on the doses provided, an evaluation of 2.2/4g per day for humans is quite high, re-evaluating with the actual amount ingested could reduce this figure.
In nutritional experiment is difficult to control the exact dose individually. We normally provide the nutrient in the diet and an average intake is assumed by all the animals if they are housed in groups, or you can control food intake if they are housed individually and express the dose per kg/BW. In the case on mice in the intelliCage, we were able to control that all the water was consumed. The cage has four corners with two water bottles per corner. The system is program to open the door when the transponders of the animals are detected. So then three mice were allocated per corner which reduced competence for water intake. In this way, the variability of water intake and dose was minimized. In the case of the rats, they were house individually and the gelatin was offered twice. The animals were trained to take the gelatin, in fact, they loved it and most of the time they took all the portion. The dose between 2.2-4 g looks high at first sight but the most study dose of Ca HMB was 3g/day.
3. The fact that mice were not in isolation is very original and interesting; just for curiosity, it would be interesting to compare with isolated animals.
Behaviour is for sure influence by housing conditions. Usually, performance is improved in group because animals can lean from each other. However, it is unusual in mice and it would not be practical when using the IntelliCage. The cage is big for only one mouse and it is design to test animals in groups.
4. Lines 182 to 205 are difficult to follow
We can see the reviewer’s point. It is a mix of statistical jargon and the DMTP task. It means that the change on delay was decided every day on a statistical analysis of the slope of the regression line made with the percentage of correct results of each day and 5 days before. If there was no slope (the slope was not significantly different from 0) then the plateau was reached and there was no more learning. Then the delay was increased, and the next phase was started. We simplified this part and separated it into two paragraphs to improve readability.
5. What is the rationale for the doses? Why are the doses so high for the mice?
The rational was based on previous experiments in rats for cognition (references 18-20) and for body composition and muscle performance (reference 22).The doses in mice were approximately equivalent to rat doses taking into consideration extrapolation factors between species according to differences in body surface area (reference 38). The conversion factor was doble in mice than in rats. Internal pharmacokinetic data also showed that plasma concentration in mice after and acute dose was half than in rats.